# The Role of the Nrf2 Signaling in Obesity and Insulin Resistance

**DOI:** 10.3390/ijms21186973

**Published:** 2020-09-22

**Authors:** Shiri Li, Natsuki Eguchi, Hien Lau, Hirohito Ichii

**Affiliations:** Department of Surgery, University of California, Irvine, CA 92868, USA; neguchi@hs.uci.edu (N.E.); hlau2@uci.edu (H.L.)

**Keywords:** obesity, insulin resistance, Nrf2, inflammation, oxidant stress

## Abstract

Obesity, a metabolic disorder characterized by excessive accumulation of adipose tissue, has globally become an increasingly prevalent disease. Extensive studies have been conducted to elucidate the underlying mechanism of the development of obesity. In particular, the close association of inflammation and oxidative stress with obesity has become increasingly evident. Obesity has been shown to exhibit augmented levels of circulating proinflammatory cytokines, which have been associated with the activation of pathways linked with inflammation-induced insulin resistance, a major pathological component of obesity and several other metabolic disorders. Oxidative stress, in addition to its role in stimulating adipose differentiation, which directly triggers obesity, is considered to feed into this pathway, further aggravating insulin resistance. Nuclear factor E2 related factor 2 (Nrf2) is a basic leucine zipper transcription factor that is activated in response to inflammation and oxidative stress, and responds by increasing antioxidant transcription levels. Therefore, Nrf2 has emerged as a critical new target for combating insulin resistance and subsequently, obesity. However, the effects of Nrf2 on insulin resistance and obesity are controversial. This review focuses on the current state of research on the interplay of inflammation and oxidative stress in obesity, the role of the Nrf2 pathway in obesity and insulin resistance, and the potential use of Nrf2 activators for the treatment of insulin resistance.

## 1. Introduction

Obesity, characterized by excessive storage of fatty acids in adipose tissue, is dramatically increasing and the global obesity epidemic is worsening [1,2]. Obesity is strongly associated with diabetes mellitus, stroke, hyperlipoidemia, cancers, cardiovascular disease, and osteoarthritis [3]. Type 2 diabetes, one of the most widespread metabolic diseases, and obesity are tightly linked due to their association with insulin resistance. As a state of chronic inflammation, obesity is a well-known risk factor for the onset and development of insulin resistance [4]. Additionally, inflammation and oxidative stress, which are considered to pathologically contribute to obesity, have been demonstrated to play important roles in the development of insulin resistance [5]. Currently, insulin resistance is a key therapeutic target for obese patients with type 2 diabetes. Therefore, understanding the interplay of obesity, inflammation, and oxidative stress in the pathogenesis of insulin resistance is a critical area of study.

Nuclear factor erythroid 2 related factor 2 (Nrf2) is a basic leucine transcription factor that plays a crucial role in the maintenance of redox and metabolic homeostasis by regulating cellular antioxidants and decreasing inflammatory stress (Figure 1) [6,7]. In the last two decades, Nrf2 has been extensively studied in multiple disorders, including insulin resistance, due to its strong capacity to regulate the expression of downstream target antioxidant protein [7,8,9]. However, activation of Nrf2-ARE signaling can lead to either beneficial or harmful effects depending on the diseases and processes of the disease [6]. For example, activation of Nrf2 may have a protective effect against oxidative stress for diseases related to chronic inflammation and reactive oxygen species (ROS) production; however, prolonged activation of Nrf2 results in metabolic changes that result in reductive stress and ultimately, contribute to disease progression [10]. Since obesity has been closely correlated with inflammation and oxidative stress, the potential protective role of the Nrf2 pathway is of great interest [5]. In order to better understand and elucidate the complex and controversial effects of the Nrf2 pathway on the pathogenesis of obesity, this review will examine and discuss the current state of research on insulin resistance and the Nrf2 pathway in obesity.

## 2. Inflammation and Oxidative Stress in Obesity

Obesity is the accumulation of abnormal or excessive adipose tissue, usually caused by increased energy consumption and reduced physical exercise. Adipose tissue not only functions as an energy storage organ, but also exerts important endocrine roles in the body through the secretion of biologically active compounds that regulate metabolic homeostasis [11]. The first discovery established by Zhang, Y. et al. identified that leptin, a peptide hormone primarily secreted by adipocyte, is essential for body weight control and other biological functions [12]. Besides adipocytes, adipose tissue also contains various types of cells such as immune cells, fibroblasts, and endothelial cells, which secrete different types of hormones, such as peptide hormones (adipokines) and bioactive lipids (lipokines), to regulate appetite, glucose, and lipid metabolism [13,14]. In a healthy state, endocrine factors secreted by adipose tissue maintain organs and metabolic homeostasis. However, in obesity, the proinflammatory profiles with abnormal secretion of adipokines and lipokines of adipocytes and immune cells in adipose tissue contribute to the initiation of chronic inflammation and oxidative stress [15,16]. The close association and interplay between obesity, inflammation, and oxidative stress have been highlighted in several studies [17,18].

Many studies have demonstrated that obesity is associated with low grade chronic systemic inflammation [17]. Firstly, the presence of excessive adipose tissue in obese subjects leads to secretion of inflammatory cytokines such as interleukin 6 (IL-6), tumor necrosis factor (TNF-α), monocyte chemoattractant 1 (MCP-1), interleukin 1 (IL-1), and plasminogen activator inhibitor-1 (PAI-1), critical cytokines that influence the inflammatory response [19]. Inflammation, a part of the immune response, is a complex physiological response with an ordered sequence of events that plays a crucial role in restoring tissue and organ homeostasis that was altered by harmful stimuli. The timely inflammatory response is essential to repair damaged tissue, heal wounds, and defend the body against infections [20]. However, overreaction of inflammatory response may ignite deleterious effects leading to multiple chronic disorders such as arthritis, asthma, atherosclerosis, cancer, and diabetes [11]. Two important causes for increased proinflammatory cytokine release in obesity are highlighted below: inflammatory response induced by hypoxic conditions of adipose tissue in obesity and activation of proinflammatory macrophage in obesity.

Hyperplasia and hypertrophy of adipocytes are the natural responses to stimulation of extra nutrients in obesity. In obesity, the blood flow to adipose tissue is not increased relatively to lean individuals [21,22,23]. With rapid expansion of adipose tissue in obesity, the blood supply to adipocytes may become insufficient, which leads to hypoxia [21,22,23]. Hypoxia is proposed to be one of the major triggers for adipose tissue remodeling including adipocyte death and inflammatory response in obesity [19,24,25].

Macrophage are primary innate immune cells that are present in most tissue. Besides phagocytosis, macrophages play a critical role in both innate immunity and adaptive immunity. Under exposure to inflammatory stimuli, macrophages secrete cytokines such as TNF, IL-1, IL-6, IL-8, and IL-12 [26]. It has been demonstrated that macrophages under obese conditions are another main cause of low-grade inflammation [27]. The macrophages in adipose tissue are bone marrow-derived and the number of macrophages is strongly correlated to body mass index and total body fat [28]. Adipose tissue is colonized by proinflammatory macrophages in obesity and obesity is closely associated with increases in the macrophage content of adipose tissue [29,30,31]. In obesity, circulating levels of proinflammatory cytokines, including TNF-α and IL-6, are elevated and the main sources of these inflammatory mediators are hepatic and white adipose macrophages [32,33]. The proinflammatory cytokine TNF-α in particular exerts a major influence on inflammatory response, lipid metabolism, insulin signaling, and oxidative stress. In addition, TNF-α has been shown to induce ROS production through activation of the nuclear factor B κ kinase subunit β (NF-κB) pathway, providing one of the possible links between inflammation and oxidative stress in obesity [16].

Secondly, the complex interplay of oxidative stress with inflammation and obesity has also been addressed in several studies. First off, oxidative stress is caused by an imbalance between production and accumulation of ROS in cells/tissues and the ability of a biological system to detoxify these reactive products [34]. In fact, oxidative stress may have both positive and negative effects on human health. Although ROS plays negative roles at high concentrations, at lower concentrations, ROS exerts beneficial effects regulating cell signaling cascades [35]. Furthermore, although ROS contributes to cell proliferation and metastasis in cancer cells, increased ROS levels may in turn cause oxidative stress-induced apoptosis and therefore, may give rise to a different cancer treatment method focused on enhancing ROS production in cancer cells [36]. Increasing evidence shows the strong relationship between oxidative stress and obesity. On the one hand, obesity can induce systemic oxidative stress through several molecular signaling pathways, such as superoxide generation from NADPH oxidases (NOX), oxidative phosphorylation, glyceraldehyde auto-oxidation, protein kinase C (PKC) activation, and polyol and hexosamine pathways [37,38]. There are multiple harmful conditions in obesity that could generate oxidative stress, such as hyperglycemia, elevated lipid levels, vitamin and mineral deficiencies, hyperleptinemia, endothelial dysfunction, and impaired mitochondrial function, etc. [39]. On the other hand, oxidative stress plays an important role in the pathogenesis of obesity and the development of co-morbidities [37,39]. Oxidative stress could trigger obesity by activating adipocyte differentiation, preadipocyte proliferation, and increasing the size of mature adipocytes [40,41,42]. In addition, oxidative stress could alter food intake by impacting hypothalamic neurons that control satiety and hunger behavior [43]. Equally important to the role of oxidative stress in pathogenesis of obesity is its pathogenic role in chronic inflammatory disease. There are extensive studies indicating that oxidative stress plays a pathogenic role in chronic inflammatory diseases [44]. Proinflammatory cytokines, such as TNFα, IL-1, and IL-6, have been suggested to contribute to oxidative stress-induced inflammation, and continued oxidative stress/inflammation may contribute to cancer, diabetes, and cardiovascular and neurological disorders [38]. Oxidative stress activates a variety of transcription factors including NF-κB, activator protein 1 (AP-1), tumor protein p53 (p53), hypoxia-inducible factor 1α (HIF-1α), peroxisome proliferator-activated receptor γ (PPAR-γ), β-catenin/Wnt, and Nrf2, which can lead to the expression of numerous targeted genes, including proinflammatory cytokines and chemokines [44].

## 3. Insulin Signaling and Insulin Resistance

Insulin is a pleiotropic hormone produced by β-cell of pancreatic islets, which regulates the metabolism of carbohydrates, fats, and protein by inducing the uptake of glucose from blood and upregulating protein synthesis in the muscle, promoting glucose utilization and triglyceride synthesis while inhibiting glucose production in the liver, and promoting glucose and fatty acid uptake and inhibiting lipolysis in the adipose tissue [45]. These effects of insulin are mediated by its signal transduction pathway involving the insulin receptor and insulin like growth factor 1 receptor on the cell membrane [46]. Binding of insulin on the insulin receptor or insulin like growth factor 1 receptor (IGF1R) results in the phosphorylation of insulin receptor substrate 1/2 (IRS1/2) at its tyrosine residues, which results in the subsequent activation of two main pathways, the phosphoinositide3-kinase(PI3K)/protein kinase B (AKT) pathway and the mitogen-activated protein kinase (MAPK) pathway [46,47].

Insulin resistance is a pathologic condition in which insulin-dependent cells, such as skeletal muscle, liver, and adipocytes stop responding properly to normal circulatory levels of insulin [48]. In skeletal muscle, insulin promotes glucose uptake by stimulating translocation of the glucose transporter 4 (GLUT4) to the plasma membrane, and impaired skeletal muscle insulin signaling results in decreased glucose removal from the blood. In the liver, insulin inhibits the expression of key gluconeogenic enzymes; therefore, insulin resistance leads to elevated hepatic glucose production. In adipose tissue, levels of the insulin-regulated GLUT4 are reduced in insulin-resistance obesity. Moreover, insulin decreases lipase activity and inhibits free fatty acid efflux out of adipocytes [49,50].

## 4. Inflammation and Oxidative Stress Cause Insulin Resistance in Obesity

Although the precise underlying mechanisms of insulin resistance are still unclear, there is increasing evidence suggesting that inflammation and oxidative stress are mainly involved in the pathogenesis of insulin resistance [51]. Inflammation and insulin resistance may be closely linked together by proinflammatory cytokines and chemokines. The infiltration of inflammatory cells into adipose tissues leads to the increased secretion of proinflammatory cytokines and chemokines such as TNFα, MCP-1, C reactive protein (CRP), and interleukins, and these same inflammatory mediators have been shown to be upregulated in insulin resistance [52]. Interestingly, overexpression of MCP-1, a chemokine secreted primarily by macrophage and endothelial cells which helps to recruit monocytes to damaged and inflamed tissue, in adipose tissue, resulted in increased macrophage infiltration and insulin resistance [53,54]. Moreover, other obesity-induced inflammatory mediators such as TNFα, IL-1 family, and IL-6 have also exhibited the ability to promote insulin resistance [33,52,55]. Several signaling pathways, including IκB kinase (IKKβ)/NF-κB and c-Jun *N*-terminal kinase (JNK) pathways, which are activated in adipose tissues and the liver, may play an important role in inflammation-induced insulin resistance [50,56,57,58]. These signaling pathways, which may be activated by TNFα, free fatty acids, ROS, and hypoxia, cause serine kinase phosphorylation of IRS-1 or IRS-2, which blocks insulin signaling and subsequently, causes insulin resistance [59]. Further, it has been shown that TNFα inhibits IRS-1 through stimulation of p55 TNF receptor [60], and activation of IKKβ and JNK1 [61,62]. Therefore, IKKβ/NF-κB and JNK pathways are important mediator pathways between inflammation and insulin resistance, and thus, could potentially be a site for intervention.

Adipose tissue macrophage has also emerged as an important player in insulin resistance. Based on the activation state and functions, macrophages are divided classically into activated macrophages (M1) and alternatively activated macrophages (M2) [63]. The M1 macrophage may promote the development of insulin resistance through secretion of proinflammatory cytokine such as TNF-α, IL-6, and IL-1b in response to stimulation by INF-γ. In contrast, M2 macrophages produce anti-inflammatory cytokines such as IL-10 and IL-4 to sustain insulin resistance [64]. M1 and M2 in adipose tissues play an interactive role in the development of insulin resistance. For instance, during the development of obesity, adipose tissue macrophages could shift from the M2 polarized state to M1 state, which leads to chronic inflammation and insulin resistance [65].

## 5. Nrf2 Antioxidant and Anti-Inflammatory Effects

Nrf2 is a transcription factor that plays a crucial role in cellular redox homeostasis under oxidative stress, inflammation, and carcinogenic conditions, etc. [66]. As stated in the previous sections, activated immune cells generate ROS that can damage macromolecules, including DNA, and produce inflammatory mediators such as cytokines and chemokines in the pathological inflammatory state of obesity. Those cytokines and chemokines can further recruit macrophages and activate NF-κB, MAPK, and Janus kinase (JAK) signal transducer and activator of transcription (STAT) signaling pathways that are involved in the development of the classical pathway of inflammation [67]. Thus, the antioxidative properties of Nrf2 is of great interest in obesity and insulin resistance.

Nrf2 is a basic region leucine zipper transcription factor which binds to the antioxidant response element and regulates expression of genes encoding cellular antioxidant and anti-inflammatory molecules. Nrf2 has a short half-life of 10–30 min, which makes the Nrf2 basal level extremely low [68]. Under normal conditions, the transcription factor Nrf2 is held in the cytoplasm via binding with Kelch-like ECH-associated protein (Keap1), an adaptor protein for Cullin3-based ubiquitin E3 ligase. Keap1 regulates the transcriptional activity of Nrf2 through the Keap1-Cullin3 ubiquitin system. Exposure to electrophiles and ROS during oxidative stress modifies Keap1 cysteine residues [69], which leads to the release of Nrf2 [70] and subsequent accumulation of Nrf2. Nrf2 then translocates to the nucleus and activates transcription of its target genes [71,72]. Nrf2 controls the expression of key components of the antioxidant system, such as glutathione and thioredoxin, and regulates the transcription of many ROS-detoxifying enzymes such as glutathione peroxidase 2, heme oxygenase, and several glutathione S transferases. Moreover, Nrf2 supports nicotinamide adenine dinucleotide phosphate (NADPH) production through the positive regulation of the principal NADPH-generating enzymes [7]. The importance of Nrf2 in redox homeostasis is further supported by an in vivo study using Nrf2-KO cells and tissues, which demonstrated that the Keap-Nrf2 pathway regulates both mitochondrial and cytosolic ROS production through its interaction with NADPH oxidase, a major ROS producer within the cell [73].

In addition to the Keap1-Cullin3 ubiquitin system, Nrf2 activity is tightly regulated by several other factors at multiple levels. Nrf2 can be regulated by aryl hydrocarbon receptor (AhR) and NF-κB at the transcriptional level. Interestingly both AhR and NF-κB increase transcription of Nrf2 protein [6]. Keap1 is the most prominent cellular factor that controls Nrf2 protein stability at the post translational level. Besides Keap1 however, β transducing repeat-containing protein (β-TrCP)-cullin 1 (Cul1)-, HMG-CoA reductase degradation protein 1 (Hrd1)-dependent, and NF-κB pathways are the most notable. Activation of both β-TrCP-Cul1-and Hrd1-dependent pathways results in degradation of Nrf2. They are therefore negative regulators for Nrf2 protein stability [6]. NF-κB inhibits the Nrf2 pathway through the interaction of transcription factor p65 and Keap1 leading to decreased Nrf2 binding to its cognate DNA sequences and enhanced Nrf2 ubiquitination [74]. Furthermore, Nrf2 activity can also be regulated after its nuclear translocation. For example, cAMP-responsive element-binding protein (CREB) in the nucleus has been shown to interact and activate Nrf2 [75].

NF-κB is an important mediator between inflammation and insulin resistance as discussed previously and an intricate/complex interaction exists between Nrf2 and NF-κB. NF-κB is a protein complex that controls DNA transcription in almost all types of animal cells and plays a critical role in regulating the survival, activation, and differentiation of innate immune cells and the inflammatory process [76]. Under oxidative stress, activation of IkB kinase can cause the phosphorylation of IkB, therefore resulting in the release and nuclear translocation of NF-κB. NF-κB leads to the transcription of proinflammatory mediators including TNF-α, IL-6, IL-1, and iNOS. A plethora of studies have indicated that the Nrf2 and NF-κB signaling pathways interact to control the transcription or function of downstream target proteins [67]. For example, NF-κB can directly inhibit Nrf2-ARE signaling through depriving CREB binding protein from Nrf2 and promoting recruitment of histone deacetylase 3 to bZip Maf transcription factor protein (MafK) [77]. In addition, NF-κB also suppresses the Nrf2 pathway by interacting nuclear factor P65 with Keap1 [74]. On the other hand, Nfr2 negatively regulates the NF-κB signaling pathway by increasing antioxidant defenses and Nrf2-dependent downstream HO-1 expression [78,79,80].

Besides regulation of antioxidant systems, in inflammatory conditions, activated Nrf2 exerts its anti-inflammatory and antioxidant properties through inhibiting proinflammatory cytokines, including IL-6 and IL-1B, and downregulating the production of IL-17, T helper type 1(Th1), and T helper type 17 (Th17) [81,82]. A recent study by Kobayashi and the colleagues proved that Nrf2 may also act as an anti-inflammatory regulator in a ROS-independent manner, which indicates that both inflammation and oxidative stress could independently be the triggers of cytoprotective pathways of Nrf2 [81].

## 6. The Role of Nrf2 Pathway in Obesity and Insulin Resistance

Due to the crucial role of anti-inflammatory and antioxidant defense played by the Nrf2 pathway in metabolic homeostasis, there are numerous studies performed to investigate the effect and mechanisms of the Nrf2 pathway in obesity and insulin resistance. In animal models, two different approaches have been frequently used to investigate the regulation of Nrf2: genetic manipulation such as Nrf2 Knockout (KO) or Keap1 Knockdown (KD) mice and administration of Nrf2 pharmacological activators [83,84,85,86,87,88,89,90,91,92,93,94,95,96]. Nrf2 KO mice lack constitutive expression of Nrf2 and hence, exhibit an inability to induce cytoprotective genes under oxidative stress. In contrast to Nrf2 KO mice, Keap1 KD mice are genetically engineered to investigate the effects of constitutive overexpression of Nrf2. Interestingly, some conflicting results have been reported based on the different approaches and animal models. Here, we focus on and discuss the recent state of research on the role of Nrf2 in obesity and insulin resistance.

### 6.1. The Role of Nrf2 in Obesity

Chartoumpekis DV and colleagues investigated the role of Nrf2 in a long-term (180 days) high-fat diet-induced (60 kcal% fat) obese mouse model [83]. Their study showed that Nrf2-KO mice were partially protected from high-fat diet-induced obesity. In addition, the levels of fibroblast growth factor 21 (FGF21) in the plasma, and FGF21 mRNA in liver and white adipose tissue in Nrf2-KO mice were significantly higher compared to WT mice. In another study conducted on high-fat diet (60 kcal% fat)-induced obese mice, Shin S. et al. showed that Nrf2-disrupted mice gained less weight than WT mice under a high fat diet [84]. Consistent with these studies, which exemplify negative regulation of high-fat diet-induced weight gain in Nrf2-KO mice, Pi J. et al. showed that targeted knockout of Nrf2 in mice decreases adipose tissue mass and protects against weight gain under a high-fat diet (41 kcal% fat) [85]. Interestingly, keap1-hypo mice exhibited similar results as Nrf2 KO mice under a high fat diet. Slocum et al. demonstrated that a Keap1-hypo mouse model with Nrf2 pathway activation suppressed high-fat diet-induced (60 kcal% fat) obesity and decreased deposition of lipids and cholesterol in the liver [86]. Moreover, Xu J. et al. found that high-fat diet-induced (60 kcal% fat) obesity and lipid accumulation in white adipose tissue was decreased in Keap1-knockdown mice with genetically enhanced Nrf2 activity [87]. These studies using Nrf2-KO or Keap1-knowdown mice show that the Nrf2 pathway plays an important role against weight gain under high-fat diet feeding.

In contradiction, Liu et al. reported that when fed a high-fat diet (22 kcal% fat) for 8 weeks, Nrf2-KO mice gained slightly more weight than the WT mice without significant difference [74]. In another study, Zhang Y and colleagues investigated the effect of the Nrf2 pathway on high-fat diet-induced (39.7 kcal% fat) obesity among Nrf2-KO, WT, and Keap1-knockdown mice, which concluded that the genetic alteration of Nrf2 does not prevent diet-induced obesity in mice [88].

In summary, most studies using mice with Nrf2-deficiency or increased Nrf2 activity showed reduced body weight compared to WT mice under a high-fat diet (60 kcal% fat) condition. Two studies [88,97] using high-fat diets with relatively low fat content (22 and 39.7 kcal% fat, respectively), showed no significant changes in weight gain between Nrf-KO mice and WT mice. This may indicate that diet composition (especially fat content) may in part be responsible for the conflicting results.

With increasing evidence showing that Nrf2 plays a key role in the regulation of energy metabolism, various Nrf2 pharmacological activators have been tested in high-fat diet-induced obese rodent models for possible clinical application. Oltipraz, a potent Nrf2 activator, can prevent body weight and fat gain induced by high-fat diet [89]. Moreover, Shin S. et al. demonstrated that treatment with CDDO imidazolide, a potent Nrf2 activator, could prevent high-fat diet-induced increases in body weight, adipose mass, and hepatic lipid accumulation via increasing oxygen consumption and energy expenditure, and decreasing food intake in wild-type mice, but not in Nrf2-disrupted mice, which indicates that activation of the Nrf2 pathway prevents obesity [84]. Lastly, parthenolide and epigallocatechin 3-gallate can inhibit obesity and obesity-induced inflammatory responses via Nrf2/Keap1 signaling under high-fat diet conditions [90,91].

### 6.2. The Role of Nrf2 in Insulin Resistance

Due to the pivotal cytoprotective, antioxidant, and anti-inflammatory role played by the Nrf2 pathway, a number of studies have been conducted to investigate its effect on insulin resistance and underlying mechanisms. Genetically modified mouse models such as Nrf2 KO and Keap1 KD have been frequently used in the setting of high-fat diet-induced obesity because of the strong association between obesity and insulin resistance.

In Nrf2 KO mice, lack of Nrf2-induced cytoprotection and antioxidants is expected to aggravate insulin resistance under high-fat diet conditions. A study conducted by Liu et al. demonstrated that Nrf2 deficiency can induce hepatic insulin resistance by activation of the NF-κB signaling pathway in mice fed a high-fat diet. In addition, malondialdehyde, an indicator of oxidative stress, is increased and the level of glutathione is decreased in Nrf2-KO mice [97]. However, in contrast to the expectation and study by Liu et al., most studies found that Nrf2 deficiency has a positive effect on insulin resistance and glucose homeostasis in the high-fat diet-induced obese mouse model. Similar to the results on obesity mentioned previously, the deletion of Nrf2 ameliorates insulin resistance compared to WT mice under high-fat diet feeding [83,85]. Moreover, Nrf2 KO mice exhibited better insulin sensitivity than WT mice when fed a high-fat diet [98]. Finally, both in in vitro and in vivo settings, Meher A.K. et al. demonstrated that Nrf2 deficiency protects mice from insulin resistance in long-term high-fat diet feeding by decreasing adipose tissue inflammation [92]. It is noteworthy to mention that these studies, which showed a positive effect of the deletion of Nrf2 on insulin resistance, used systemic Nrf2 KO mice rather than tissue-specific deletion [83,85,92,98]. A more recent study used target-specific knockout Nrf2 mice to further investigate the specific effect of Nrf2 on individual metabolic organs [93]. This study indicated that high-fat diet-fed mice with cell-specific deletion of Nrf2 in adipocytes showed deteriorated insulin resistance; in contrast, mice with deletion of Nrf2 in hepatocytes showed improved insulin sensitivity [93]. Similarly, a study using leptin-deficient ob/ob mice showed that specific ablation of the Nrf2 gene in adipocytes led to reduced white adipose tissue mass, but resulted in more severe metabolic syndrome with aggravated insulin resistance. The study revealed that Nrf2 in adipocytes plays a novel role in improving insulin resistance by upregulating antioxidant gene expression that leads to a decrease in cellular ROS [94]. Therefore, as evidenced from the discussed studies, the complicated nature of Nrf2 pathway has produced inconsistent results on the effect of Nrf2 on obesity and insulin resistance (Table 1). 

In Keap1-deficient mice, a constitutive increased activity of Nrf2 provides an efficient genetically modified tool to investigate the role of Nrf2 in various disorders. The same study done by Xu J. et al. demonstrated that increased Nrf2 activity in Keap1-KD leptin-deficient ob/ob mice inhibits insulin signaling and aggravates insulin resistance under a short-term high-fat diet feeding [87]. On the contrary, by generating a skeletal muscle-specific Keap1 knockout lean mouse model, Uruno A. et al. demonstrated that increasing Nrf2 activation in skeletal muscle improves insulin resistance via increasing glycogen branching enzyme and phosphorylase b kinase α subunit protein expression [95]. Constitutively enhanced Nrf2 signaling induced by knocking down Keap1 prevents impaired metabolism and insulin resistance in lipodystrophic mice via inhibition of lipogenic enzymes in the liver [96].

The exact mechanisms underlying Nrf2′s role in insulin resistance have not been fully clarified. The inconsistent results on the role of Nrf2 in insulin resistance from these studies using Nrf2 KO and Keap1 KD mouse models suggest that deletion or activation of Nrf2 in different specific tissues may lead to different effects on insulin resistance (Table 2). Moreover, the difference in length of high-fat diet feeding, obesity severity in mice, and animal genetic background may also exert influence on insulin resistance. Instead of whole-body deletion of Nrf2 or knocking down Keap1, more future studies focusing on specific targeted tissue deletion or activation of Nrf2 are needed.

Insulin resistance may be a result of reduced antioxidant capacity evident in obesity [99]. Ozata M. et al. also showed that the activity of antioxidant enzymes, such as SOD and GPX in obese patients, is significantly decreased when compared to healthy individuals [100]. In addition, an animal study also demonstrated that antioxidant activity is significantly decreased in obese rats compared to lean rats [101]. Pharmacological Nrf2 activator is therefore expected to ameliorate insulin resistance by counteracting obesity-induced oxidative stress and inflammation and maintaining oxidant and antioxidant homeostasis. Cheng A. et al. confirmed that activation of Nrf2 by resveratrol attenuated methylglyoxal-induced insulin resistance in Hep G2 cells through activating the ERK pathway but not through the p38 or JNK pathways, which leads to elevated levels of HO-1 and glyoxalase expression [102]. Moreover, other potent Nrf2 activators, such as oltipraz, curcumin, and notoginsenoside R1 have been investigated in obese mice. The study by Yu Z et al. demonstrated that oltipraz administration can upregulate Nrf2 target antioxidant enzymes, such as SOD and HO-1, and suppress inflammation and ER stress in high-fat diet-induced obese mice [89]. Similarly, curcumin, a natural phytocompound, can dramatically reverse high-fat diet-induced elevation of malondialdehyde (MDA) and ROS and decrease HO-1 expression in skeletal muscle, which results in improved insulin resistance in obese mice [103]. Lastly, the recent study by Zhai Y. et al. showed that administration of notoginsenoside R1, one major component of Panax notoginseng, significantly improved insulin resistance and dyslipidemia in diabetic mice [104].

## 7. Conclusions

Obesity is highly associated with the development of insulin resistance. Multiple factors including inflammation, oxidative stress, hormones, lipids, glycerol, etc., are involved in the development of insulin resistance. This review focused specifically on the role of inflammation and oxidative stress generated by adipose tissues in the pathogenesis of insulin resistance. Of note, it was discussed that the infiltration of inflammatory cells into adipose tissues results in the release of proinflammatory cytokines, which subsequently leads to inflammation-induced insulin resistance. Oxidative stress is considered to be an important factor that triggers this cascade of events in adipose tissue, which leads to the development of inflammation-induced insulin resistance. The Nrf2 molecular pathway has been demonstrated to be crucial for suppressing inflammation and oxidative stress and restoring tissue and organ homeostasis. Some of the studies on the role of Nrf2 in obesity and insulin resistance are contradictory and therefore, require more research studying the tissue-specific effects of Nrf2 KO and Nrf2 overexpression. However, the importance of Nrf2 in obesity and insulin resistance is clearly evident and the potential use of Nrf2 activator as a treatment method will continue to be an exciting area to advance. In this review, we discussed the role of Nrf2 in obesity with a focus on the anti-inflammatory and antioxidant effects of the Nrf2 pathway on insulin resistance.

## Figures and Tables

**Figure 1 ijms-21-06973-f001:**
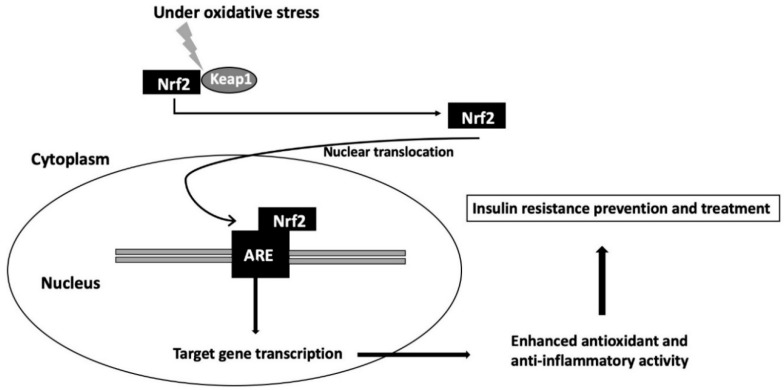
Nrf2 signaling pathway. Nrf2 bound to Keap1 in cytoplasm during low stress states is activated by oxidative stress, leading to dissociation from Keap1 and translocation to the nucleus. Nrf2 then binds to ARE that leads to cytoprotective antioxidant gene transcription. As a result, enhanced antioxidant and anti-inflammatory activity can prevent or treat insulin resistance.

**Table 1 ijms-21-06973-t001:** The role of Nrf2 pathway in obesity and insulin resistance.

Model	Obesity	Insulin Resistance	Authors	Year	Reference
Nrf2 inactivation					
95 days HFD (60 kcal% fat) in Nrf2 disrupted mice	Decreased	N/A	Shin, S. et al.	2009	[84]
12 weeks HFD (41 Kcal% fat) in Nrf2-KO mice	Decreased	Decreased	Pi, J. et al.	2010	[85]
180 days HFD (60 kcal% fat) Nrf2 KO mice	Decreased	Decreased	Chartoumpekis, D.V. et al.	2011	[83]
10 weeks HFD(60 kcal% fat) in Nrf2 KO mice	Decreased	Decreased	Meher A.K. et al.	2012	[92]
adipocyte-specific Nrf2 KO in Ob/Ob mice	Decreased	Increased	Xue, P. et al.	2013	[94]
170 days HFD(60 kcal% fat) in adipocyte-specific Nrf2 KO mice	No change	Increased	Chartoumpekis, D.V. et al.	2018	[93]
170 days HFD(60 kcal% fat) in hepatocyte-specific Nrf2 KO mice	No change	Decreased	Chartoumpekis, D.V. et al.	2018	[93]
12 weeks HFD (39.7 kcal% fat) Nrf2 KO mice	No change	N/A	Zhang, Y.K. et al.	2012	[88]
20 weeks HFD (45 kcal% fat) Nrf2 KO mice	Decreased	Decreased	Meakin, P.J. et al.	2014	[98]
8 weeks HFD (22 kcal% fat) Nrf2 KO mice	Slight increase (not significant)	Increased	Liu Z. et al.	2016	[97]
Nrf2 activation					
95 days HFD (60 kcal% fat) in female mice treated with Nrf2 activator CDDO-Im	Decreased	N/A	Shin, S. et al.	2009	[84]
28 weeks HFD (45 Kcal% fat) in mice treated with Nrf2 activator oltipraz	Decreased	Decreased	Yu, Z. et al.	2011	[89]
17 weeks HFD (45 kcal% fat)in mice with epigallocatechin 3-gallate	Decreased	N/A	Sampath, C. et al.	2017	[90]
90 days HFD (60 kcal% fat) in Keap1 hypo mice	Decreased	N/A	Slocum, S.L. et al.	2016	[86]
36 days HFD(60 kcal% fat)Lep(ob/ob) Keap1 knockdown (KD) mice	Decreased	Increased	Xu J. et al.	2012	[87]
12 weeks HFD (39.7 kcal% fat) Keap1 KD mice	No change	N/A	Zhang, Y.K. et al.	2012	[88]
Keap1 KD in lipodystrophic mice (achieved through overexpression of Notch)	N/A	Decreased	Chartoumpekis, D. V.	2018	[96]
skeletal muscle-specific Keap1 KO mice	Decreased	Decreased	Uruno, A. et al.	2016	[95]

**Table 2 ijms-21-06973-t002:** The role of Nrf2 pathway in insulin resistance: systemic vs. tissue-specific KO of Nrf2.

Model	Systemic or Tissue-Specific KO	Obesity	Insulin Resistance	Authors	Year	Reference
Insulin Resistance Decreased						
Nrf2 Inactivation						
12 weeks HFD (41 kcal% fat) in Nrf2-KO mice	Systemic	Decreased	Decreased	Pi, J. et al.	2010	[85]
180 days HFD (60 kcal% fat) Nrf2 KO mice	Systemic	Decreased	Decreased	Chartoumpekis, D.V. et al.	2011	[83]
10 weeks HFD(60 kcal% fat) in Nrf2 KO mice	Systemic	Decreased	Decreased	Meher A.K. et al.	2012	[92]
170 days HFD(60 kcal% fat) in hepatocyte-specific Nrf2 KO mice	Hepatocyte	No change	Decreased	Chartoumpekis, D.V. et al.	2018	[93]
20 weeks HFD (45 kcal% fat) Nrf2 KO mice	Systemic	Decreased	Decreased	Meakin, P.J. et al.	2014	[98]
Nrf2 Activation						
28 weeks HFD (45 Kcal% fat) in mice treated with Nrf2 activator oltipraz	Systemic	Decreased	Decreased	Yu, Z. et al.	2011	[89]
Keap1-KD in lipodystrophic mice (achieved through overexpression of Notch)	Systemic	N/A	Decreased	Chartoumpekis, D. V. et al.	2018	[96]
skeletal muscle-specific Keap1 KO mice	Skeletal Muscle	Decreased	Decreased	Uruno, A. et al.	2016	[95]
Insulin Resistance Increased						
Nrf2 Inactivation						
adipocyte-specific Nrf2 knockout in Ob/Ob mice	Adipocyte	Decreased	Increased	Xue, P. et al.	2013	[94]
170 days HFD(60 kcal% fat) in adipocyte-specific Nrf2 KO mice	Adipocyte	No change	Increased	Chartoumpekis, D.V. et al.	2018	[93]
8 weeks HFD (22 kcal% fat) Nrf2 KO mice	Systemic	Slight increase (not significant)	Increased	Liu Z. et al.	2016	[97]
Nrf2 activation						
36 days HFD(60 kcal% fat) Lep(ob/ob)-Keap1-knockdown (KD) mice	Systemic	Decreased	Increased	Xu J. et al.	2012	[87]

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
