# Peer review of "The Role of the Nrf2 Signaling in Obesity and Insulin Resistance"

_ijms, 2020, doi:10.3390/ijms21186973_

Round 1
Reviewer 1 Report
In the present review, Li et al. provide an analysis of the role of the transcription factor Nrf2 focusing their work on obesity and insulin resistance.
Although, as properly stated by the authors, the topic is of primary importance for public health, it has to be taken in account that other papers fully or at least in part, address this subject, such as Seo and Lee, Oxid Med Cell Longev. 2013, Zhang et al., Rev Endocr Metab Disord. 2015 and the very recent Vasileva et al., Pharmacol Res. 2020.
In general, Section 1-3 are fragmented, and a clear focus is missing. Too many topics are listed, and barely sketched; a better connection between them and a more in-depth explanation is needed. For example, section 2 explain the main lines of insulin signaling and its effects, but almost nothing is told about insulin resistance. Another example is in section 1 (line74-85) and 3 (141-149): the reason why the authors extensively describe macrophages role is unclear, as no correspondence between them and the role of Nrf2 in obesity and insulin resistance is found subsequently. The authors could consider to exclude from sections 1-3, the topics that are not related with the role of Nrf2 in obesity and insulin resistance and to expand the ones that are more related with the issue they are trying to address. Sections 5.1 is confusing, probably due to the conflicting results that the authors are presenting and should be reorganized, as a proper introduction and a discussion to a possible role of Nrf2 in obesity is missing.
1) Many references in the text should be added, in order to clarify and provide hints for interpretation of the statements made by the authors. For example:
Line 32: “Additionally, inflammation and oxidative stress, which are considered to pathologically contribute to the obesity, have also been demonstrated to play important roles in the development of insulin resistance.”
Line 56: “Many studies demonstrated that obesity is associated with low-grade chronic systemic inflammation”.
Line 77: It has been demonstrated that macrophage under obese conditions is another main cause of low-grade inflammation.
2) in line 42: “activation of Nrf2-ARE signaling can lead to either beneficial or harmful effects depending on the diseases and processes of the disease”: The authors could expand this with a couple of examples, to better clarify what they are meaning (e.g. cancer vs neurodegenerative diseases?)
3) The aim of the review (line 44) could be better elucidated, for example, highlighting the importance that Nrf2 signaling could have in this context
4) In line 86: the “Oxidative stress, which plays both bad and good roles in human health”, this assumption need to be better clarified. If the authors are considering oxidative stress as “the imbalance between production and accumulation of oxygen reactive species (ROS) in cells and tissues and the ability of a biological system to detoxify these reactive products” (line 87-88), this, in general, is considered a pathologic condition involved in the development or being a secondary disfunction of many disorders and it is not likely to have a good role in human health
5) In line 101, the authors affirm that “There are extensive studies indicating that oxidative stress plays a pathogenic role in chronic inflammatory disease and continued oxidative stress can directly induce chronic inflammation contributing to cancer, diabetes, cardiovascular and neurological disorders [38]”. Nevertheless, the cited reference “Oxidative stress, inflammation, and cancer: How are they linked?” does not account for such a statement in full.
6) Starting from line 108 there is a huge paragraph in which proper citations are missing: “Insulin is a pleiotropic hormone produced by beta cell of pancreatic islets which regulates the metabolism of carbohydrates, fats and protein by inducing the uptake of glucose from blood and up regulating protein synthesis in the muscle, promoting glucose utilization and triglyceride synthesis while inhibiting glucose production in the liver, and promoting glucose and fatty acid uptake and inhibiting lipolysis in the adipose tissue. These effects of insulin are mediated by its signal transduction pathway involving insulin receptor and insulin like growth factor 1 receptor on the cell membrane. Binding of insulin on insulin receptor or IGF1R results in the phosphorylation of insulin receptor substrate 1/2 (IRS1/2) at its tyrosine residues, which results in the subsequent activation of two main pathways, the Phosphoinositide3-kinase(PI3K)/AKT pathway and the mitogen activated protein kinase (MAPK) pathway[39]”.
7) Line 135: “Several signaling pathways, including IKKβ/NF‐κB and JNK pathways that are activated in adipose tissue and liver may play the important role in inflammation-induced insulin resistance [42, 45-47]. Those signaling pathways can be activated in obesity by TNF-α, free fatty acids, ROS and hypoxia. TNF-α inhibits IRS-1 in the insulin signaling pathway through stimulation of p55 TNF receptor [48], and activation of IKKβ and JNK1 are involved in IRS-1 inhibition by TNF-α[49, 50].”
This is very confusing and need to be better clarified. In particular, a link with the purpose of this review is not clear and, thus, it should be highlighted.
8) Line 151-155: there is no clear relationship with the title “Nrf2 antioxidant and anti-inflammatory effects”. Although the cited review (ref n°54) gives some clues to the reason why the authors made this introduction, it should be better clarified or removed.
Importantly, Nrf2 mechanism of function is just being introduced, thus, if there is a relationship between Nrf2, inflammation and the activation of the mentioned pathways it could considered to be stated below. More, proper citations for the statement done by the authors should be made, as the cited paper investigate the roles of Nrf2 in the inflammation process and not what the authors reports, that is properly cited in the ref n°54.
9) Line 156: the full name of Nrf2 has already been introduced, thus it should be removed.
10) Line 156-162. This paragraph is very fragmented and give only sparse and little information on how Nrf2 works and is regulated. Thus, it should be expanded properly. Maybe the authors could take advantage by the fact that Nrf2 regulates the transcription of 1% of the genome, not only the expression of genes encoding cellular antioxidant and anti-inflammatory response, that the Keap1-Cullin3 ubiquitin system is not the only mechanism that regulate Nrf2 activity and stability. Moreover, many of the statements regarding NRF2-NFKb interaction made by the authors just below, could be introduced here explaining how NRF2 interacts with CREB, p65.
11) Line 162: the statement “Nrf2 influences sensitivity to physiologic and pathologic processes affected by oxidative and electrophilic stress” is not clear and should be clarified
12) Line 174-178. It is not clear why the authors introduce NF-Kb signaling axis, in regard with the title of the paragraph. Although the interaction between NF-Kb and Nrf2 could be of interest, NF-Kb basic functioning seems off topic.
13) Line179-182 need to be better elucidated. For example: it is not clear what happens when NF-Kb deprives CREB binding from NRF2, because NRF2 and CREB interaction was not explained elsewhere. The same is true for the interaction with p65.
14) Line 259-261: it is not clear why “It is noteworthy to mention that these studies, which showed a positive effect of the deletion of Nrf2 oninsulin resistance, used systemic Nrf2 KO mice”, in comparison with studies in conditional Nrf2 knockout models. Are there specific advantages in the use of a model in respect to others?
15) Figures/tables to better clarify the findings summarized in 5.1 and 5.2 could be provided to help the understanding of the results reported in the most important sections of the paper.
Author Response
We thank the editor and reviewers for your thoughtful comments and positive feedback on our manuscript, which have helped us to improve the paper. We have revised the manuscript based on the reviewers’ suggestions and fixed the overlaps in the manuscript. Our point-to-point responses to individual comments are detailed below. We used the “track changes” function to highlight the changes we made in the resubmitted vision.
Reviewer 1
In the present review, Li et al. provide an analysis of the role of the transcription factor Nrf2 focusing their work on obesity and insulin resistance.
Although, as properly stated by the authors, the topic is of primary importance for public health, it has to be taken in account that other papers fully or at least in part, address this subject, such as Seo and Lee, Oxid Med Cell Longev. 2013, Zhang et al., Rev Endocr Metab Disord. 2015 and the very recent Vasileva et al., Pharmacol Res. 2020.
In general, Section 1-3 are fragmented, and a clear focus is missing. Too many topics are listed, and barely sketched; a better connection between them and a more in-depth explanation is needed. For example, section 2 explain the main lines of insulin signaling and its effects, but almost nothing is told about insulin resistance. Another example is in section 1 (line74-85) and 3 (141-149): the reason why the authors extensively describe macrophages role is unclear, as no correspondence between them and the role of Nrf2 in obesity and insulin resistance is found subsequently. The authors could consider to exclude from sections 1-3, the topics that are not related with the role of Nrf2 in obesity and insulin resistance and to expand the ones that are more related with the issue they are trying to address. Sections 5.1 is confusing, probably due to the conflicting results that the authors are presenting and should be reorganized, as a proper introduction and a discussion to a possible role of Nrf2 in obesity is missing.
We thank you for this valuable suggestion. We have changed paragraph order and added transition words and sentences in order to improve the flow of the paragraph 1-3 and made sure to highlight the interconnectedness of the discussed topics so that it appears less fragmented. Furthermore, to address the complicated nature of section 5.1 we have added a table organizing the studies.
1) Many references in the text should be added, in order to clarify and provide hints for interpretation of the statements made by the authors. For example:
Line 32: “Additionally, inflammation and oxidative stress, which are considered to pathologically contribute to the obesity, have also been demonstrated to play important roles in the development of insulin resistance.”
We thank you for bringing this our attention. We have fixed this by adding in a reference.
- Hurrle, S. and W.H. Hsu, The etiology of oxidative stress in insulin resistance. Biomed J, 2017. 40(5): p. 257-262.
Line 56: “Many studies demonstrated that obesity is associated with low-grade chronic systemic inflammation”.
We thank you for bringing this our attention. We have fixed this by adding in a reference.
- Stepien, M., et al., Obesity indices and inflammatory markers in obese non-diabetic normo- and hypertensive patients: a comparative pilot study. Lipids Health Dis, 2014. 13: p. 29.
Line 77: It has been demonstrated that macrophage under obese conditions is another main cause of low-grade inflammation.
We thank you for bringing this our attention. We have fixed this by adding in a reference.
- Lauterbach, M.A. and F.T. Wunderlich, Macrophage function in obesity-induced inflammation and insulin resistance. Pflugers Arch, 2017. 469(3-4): p. 385-396.
2) in line 42: “activation of Nrf2-ARE signaling can lead to either beneficial or harmful effects depending on the diseases and processes of the disease”: The authors could expand this with a couple of examples, to better clarify what they are meaning (e.g. cancer vs neurodegenerative diseases?)
We thank you for insightful suggestion. We have tried to better explain that Nrf2 can have both beneficial and harmful effects in disease by discussing how activation of Nrf2 may have a protective effect against oxidative stress for diseases related to chronic inflammation and ROS production, however prolonged activation of Nrf2 results in metabolic changes that result in reductive stress and ultimately contribute to disease progression
3) The aim of the review (line 44) could be better elucidated, for example, highlighting the importance that Nrf2 signaling could have in this context
We thank you for this valuable suggestion. We have tried to elucidate the importance of Nrf2 signaling in obesity and insulin resistance by connecting it to the fact that obesity and insulin resistance are both closely related to inflammation and oxidative stress, which are both pathogenic states in which Nrf2 acts on, which was described in the previous sentence.
4) In line 86: the “Oxidative stress, which plays both bad and good roles in human health”, this assumption need to be better clarified. If the authors are considering oxidative stress as “the imbalance between production and accumulation of oxygen reactive species (ROS) in cells and tissues and the ability of a biological system to detoxify these reactive products” (line 87-88), this, in general, is considered a pathologic condition involved in the development or being a secondary disfunction of many disorders and it is not likely to have a good role in human health
Thank you for your invaluable insight. We have fixed this by providing an example of when ROS may be beneficial such as when playing a role in cell signaling cascade. Additionally, we have added potential use of ROS for cancer treatment to show beneficial use of ROS for human health.
5) In line 101, the authors affirm that “There are extensive studies indicating that oxidative stress plays a pathogenic role in chronic inflammatory disease and continued oxidative stress can directly induce chronic inflammation contributing to cancer, diabetes, cardiovascular and neurological disorders [38]”. Nevertheless, the cited reference “Oxidative stress, inflammation, and cancer: How are they linked?” does not account for such a statement in full.
Thank you for your invaluable insight. We have reread the article and altered our choice of words so as to not overstate the statement. Instead of using strong words such as “directly induce”, we have changed it to “contribute to”.
6) Starting from line 108 there is a huge paragraph in which proper citations are missing: “Insulin is a pleiotropic hormone produced by beta cell of pancreatic islets which regulates the metabolism of carbohydrates, fats and protein by inducing the uptake of glucose from blood and up regulating protein synthesis in the muscle, promoting glucose utilization and triglyceride synthesis while inhibiting glucose production in the liver, and promoting glucose and fatty acid uptake and inhibiting lipolysis in the adipose tissue. These effects of insulin are mediated by its signal transduction pathway involving insulin receptor and insulin like growth factor 1 receptor on the cell membrane. Binding of insulin on insulin receptor or IGF1R results in the phosphorylation of insulin receptor substrate 1/2 (IRS1/2) at its tyrosine residues, which results in the subsequent activation of two main pathways, the Phosphoinositide3-kinase(PI3K)/AKT pathway and the mitogen activated protein kinase (MAPK) pathway[39]”.
Thank you for bringing this to our attention. We have fixed this by adding reference to each of the sentences that require citation.
- Dimitriadis, G., et al., Insulin effects in muscle and adipose tissue. Diabetes Res Clin Pract, 2011. 93 Suppl 1: p. S52-9.
- Boucher, J., A. Kleinridders, and C.R. Kahn, Insulin receptor signaling in normal and insulin-resistant states. Cold Spring Harb Perspect Biol, 2014. 6(1).
7) Line 135: “Several signaling pathways, including IKKβ/NF‐κB and JNK pathways that are activated in adipose tissue and liver may play the important role in inflammation-induced insulin resistance [42, 45-47]. Those signaling pathways can be activated in obesity by TNF-α, free fatty acids, ROS and hypoxia. TNF-α inhibits IRS-1 in the insulin signaling pathway through stimulation of p55 TNF receptor [48], and activation of IKKβ and JNK1 are involved in IRS-1 inhibition by TNF-α[49, 50].”
This is very confusing and need to be better clarified. In particular, a link with the purpose of this review is not clear and, thus, it should be highlighted.
Thank you for this insightful comment. We have changed sentence order and word choice in an attempt to better highlight the importance of JNK and IKKbeta/NF-kB pathway as a mediator pathway between inflammation and insulin resistance to further support the connection between inflammation and insulin resistance.
8) Line 151-155: there is no clear relationship with the title “Nrf2 antioxidant and anti-inflammatory effects”. Although the cited review (ref n°54) gives some clues to the reason why the authors made this introduction, it should be better clarified or removed.
Importantly, Nrf2 mechanism of function is just being introduced, thus, if there is a relationship between Nrf2, inflammation and the activation of the mentioned pathways it could considered to be stated below. More, proper citations for the statement done by the authors should be made, as the cited paper investigate the roles of Nrf2 in the inflammation process and not what the authors reports, that is properly cited in the ref n°54.
Thank you for this valuable insight. We have attempted to better connect this paragraph to the title “Nrf2 antioxidant and anti-inflammatory effects” by reemphasizing the role of Nrf2 in cellular redox homeostasis and through restating the connection between inflammation and oxidative stress.
9) Line 156: the full name of Nrf2 has already been introduced, thus it should be removed.
Thank you for bringing this to our attention. We have removed the full name of Nrf2.
10) Line 156-162. This paragraph is very fragmented and give only sparse and little information on how Nrf2 works and is regulated. Thus, it should be expanded properly. Maybe the authors could take advantage by the fact that Nrf2 regulates the transcription of 1% of the genome, not only the expression of genes encoding cellular antioxidant and anti-inflammatory response, that the Keap1-Cullin3 ubiquitin system is not the only mechanism that regulate Nrf2 activity and stability. Moreover, many of the statements regarding NRF2-NFKb interaction made by the authors just below, could be introduced here explaining how NRF2 interacts with CREB, p65.
Thank you for this insightful comment. We have discussed in more detail the regulation of Nrf2 activity by explicitly describing control at transcriptional and posttranslational level and also after Nrf2 translocation to the nucleus. We believe by discussing the regulation here, there will be less confusion when these regulatory mechanisms are addressed later on.
11) Line 162: the statement “Nrf2 influences sensitivity to physiologic and pathologic processes affected by oxidative and electrophilic stress” is not clear and should be clarified
Thank you for this valuable insight. We have removed this sentence and altered the structure of this paragraph in order to better highlight Nrf2 activation under oxidative stress which results in upregulation of antioxidant properties of the cell.
12) Line 174-178. It is not clear why the authors introduce NF-Kb signaling axis, in regard with the title of the paragraph. Although the interaction between NF-Kb and Nrf2 could be of interest, NF-Kb basic functioning seems off topic.
Thank you for this insightful comment. We added this paragraph about NF-kB and Nrf2 interaction because a study discussed later in section 5.2 addresses Nrf2 induced NF-kB activation in insulin resistance. Therefore, we believe that it is important point out how Nrf2 and NF-kB can interact and affect each other’s pathway. We have improved the paragraph discussing Nrf2 and NF-kB interaction by adding transition words and sentences to better highlight the relevance of NF-kB pathway in insulin resistance and obesity.
13) Line179-182 need to be better elucidated. For example: it is not clear what happens when NF-Kb deprives CREB binding from NRF2, because NRF2 and CREB interaction was not explained elsewhere. The same is true for the interaction with p65.
Thank you for this invaluable insight. We believe that this problem is resolved because we added a paragraph above this paragraph discussing regulation of Nrf2 pathway, which included the effect of CREB on Nrf2 activity.
14) Line 259-261: it is not clear why “It is noteworthy to mention that these studies, which showed a positive effect of the deletion of Nrf2 on insulin resistance, used systemic Nrf2 KO mice”, in comparison with studies in conditional Nrf2 knockout models. Are there specific advantages in the use of a model in respect to others?
Thank you for this valuable comment. We believe that this point is important because studies that used systemic vs specific tissue KO gave different results on the effect on insulin resistance. These differences are discussed in detail after this sentence. We have attempted to make this point clear by making a table that indicates whether the studies used systemic or specific tissue KO and what the results/effect on insulin resistance was.
15) Figures/tables to better clarify the findings summarized in 5.1 and 5.2 could be provided to help the understanding of the results reported in the most important sections of the paper.
Thank you for this valuable comment. We have made a table to organize the studies that were addressed in sections 5.1 and 5.2.
Reviewer 2 Report
The review article entitled “The role of the Nrf2 signaling in Obesity and Insulin 2 Resistance” compile the literature demonstrating the role of Nrf2 pathway in obesity and development of insulin resistance. The authors could have also focused on Leptin which is a major contributor for obesity and recognized to be a prominent obesity marker and Nrf2 link which is lacking in this review article. Although I don't have major concerns with this review, minor issues needs to be addressed appropriately.
- Introduction, subsection Inflammation and oxidative stress in obesity, line 76, elaborate abbreviations for all the cytokines used in this article
- Introduction, subsection Inflammation and oxidative stress in obesity, line 76, clarify IT-1 or IL-1?
- Introduction, subsection Inflammation and oxidative stress in obesity, line 82, TNF-a or TNF-α?
- Introduction, subsection Inflammation and oxidative stress in obesity, line 82, TNF-A or TNF-α? Maintain consistency!
- Introduction, subsection Inflammation and oxidative stress in obesity, line 87, oxygen reactive species to be changed to “ROS-reactive Oxygen Species”
- Introduction, subsection 1, Inflammation and oxidative stress in obesity, line 88, “Increasing evidence shows the relationship between oxidative stress and obesity”… add appropriate references.
- IGF1R? need expansion
- AKT? Need expansion
- GLUT4?
- Introduction, subsection 1, Inflammation and oxidative stress cause insulin resistance in obesity….line 133… TNF-a? or TNF-α
- Expansions for IKKβ/NF‐κB, JNK
- Line 144, TNF-a? or TNF-α
- Expansions for MAPK and JAK-STAT
- Line 171, In a vivo study using Nrf2-KO cells……..needs sentence rephrase
- The authors use NF-Kb and NF‐κB…….need to use one format throughout the article
- MafK? expansion
- Line 186, Th1 and Th17 needs to expand?
- Line 193,” In animal models, two 193 different approaches have been frequently used to investigate the regulation of Nrf2: manipulation such as Nrf2 Knockout or Keap-1 Knockdown mice and administration of Nrf2 195 pharmacological activators”…..provide references.
- Line 226-227, “Two studies[74] [75] using……” needs reference formatting…to be merged
- Line 245 what is Keap1 KD? is it knockdown? Need to expand KD atleast once
- The authors keep changing from Nrf2 KO to Nrf2 null mice? Need to keep one format throughout the article
- Line 257, in vitro and in vivo to be italics
Author Response
We thank the editor and reviewers for your thoughtful comments and positive feedback on our manuscript, which have helped us to improve the paper. We have revised the manuscript based on the reviewers’ suggestions and fixed the overlaps in the manuscript. Our point-to-point responses to individual comments are detailed below. We used the "track changes" function to highlight the changes we made in the resubmitted vision.
The review article entitled "The role of the Nrf2 signaling in Obesity and Insulin Resistance" compile the literature demonstrating the role of Nrf2 pathway in obesity and development of insulin resistance. The authors could have also focused on Leptin which is a major contributor for obesity and recognized to be a prominent obesity marker and Nrf2 link which is lacking in this review article. Although I don't have major concerns with this review, minor issues needs to be addressed appropriately.
1. Introduction, subsection Inflammation and oxidative stress in obesity, line 76, elaborate abbreviations for all the cytokines used in this article
Thank you for bringing this our attention. We have added the complete names of the cytokines in this sentence.
2. Introduction, subsection Inflammation and oxidative stress in obesity, line 76, clarify IT-1 or IL-1?
Thank you for bringing this our attention. We have changed IT-1 to IL-1
3. Introduction, subsection Inflammation and oxidative stress in obesity, line 82, TNF-a or TNF-α?
Thank you for bringing this our attention. We have checked and changed all the TNF- α to ensure consistency
4. Introduction, subsection Inflammation and oxidative stress in obesity, line 82, TNF-A or TNF-α? Maintain consistency!
Thank you for bringing this our attention. We have checked and changed all the TNF- α to ensure consistency
5. Introduction, subsection Inflammation and oxidative stress in obesity, line 87, oxygen reactive species to be changed to "ROS-reactive Oxygen Species"
Thank you for bringing this our attention. We have changed oxygen reactive species to reactive oxygen species
6. Introduction, subsection 1, Inflammation and oxidative stress in obesity, line 88, "Increasing evidence shows the relationship between oxidative stress and obesity"… add appropriate references.
Thank you for this insightful comment. We did not site this sentence because this general statement is supported by several references in the examples addressed after this sentence.
7. IGF1R? need expansion
Thank you for pointing this point. We have written out the expanded name of IGF-1R in the paper.
8. AKT? Need expansion
Thank you for bringing this to our attention. We have written out AKT as Protein kinase B.
9. GLUT4?
Thank you for bringing this to our attention. We have written out Glut4 as glucose transporter 4.
10. Introduction, subsection 1, Inflammation and oxidative stress cause insulin resistance in obesity….line 133… TNF-a? or TNF-α
Thank you for bringing this our attention. We have checked and changed all the TNF- α to ensure consistency
11. Expansions for IKKβ/NF‐κ, JNK
Thank you for bringing this our attention. We have written out the expanded names of each other pathways.
12. Line 144, TNF-a? or TNF-α
Thank you for bringing this our attention. We have checked and changed all the TNF- αto ensure consistency
13. Expansions for MAPK and JAK-STAT
Thank you for pointing this out. MAPK was expanded in an earlier paragraph and we have now expanded JAK-STAT.
14. Line 171, In a vivo study using Nrf2-KO cells…….needs sentence rephrase
Thank you for this insightful comment. We have rephrased and added words to the sentence to better connect it to the topic discussed in the paragraph and so that it is easier to understand.
15. The authors use NF-Kb and NF‐κ……need to use one format throughout the article
Thank you for your suggestion. We have changed all NF- κB so that it is consistent throughout the manuscript.
16. MafK? Expansion
Thank you for your suggestion. We have written out the full name of MafK in the manuscript
17. Line 186, Th1 and Th17 needs to expand?
Thank you for your suggestion. We have written out the full name of th1 and Th17.
18. Line 193," In animal models, two 193 different approaches have been frequently used to investigate the regulation of Nrf2: manipulation such as Nrf2 Knockout or Keap-1 Knockdown mice and administration of Nrf2 195 pharmacological activators"….provide references.
Thank you for your insightful comment. We have added references of studies that use Nrf2 Knockout or Keap-1 Knockdown mice and administration of Nrf2 pharmacological activators.
82.Chartoumpekis, D.V., et al., Nrf2 represses FGF21 during long-term high-fat diet-induced obesity in mice. Diabetes, 2011. 60(10): p. 2465-73.
83.Shin, S., et al., Role of Nrf2 in prevention of high-fat diet-induced obesity by synthetic triterpenoid CDDO-imidazolide. Eur J Pharmacol, 2009. 620(1-3): p. 138-44.
84.Pi, J., et al., Deficiency in the nuclear factor E2-related factor-2 transcription factor results in impaired adipogenesis and protects against diet-induced obesity. J Biol Chem, 2010. 285(12): p. 9292-300.
85.Slocum, S.L., et al., Keap1/Nrf2 pathway activation leads to a repressed hepatic gluconeogenic and lipogenic program in mice on a high-fat diet. Arch Biochem Biophys, 2016. 591: p. 57-65.
86.Xu, J., et al., Enhanced Nrf2 activity worsens insulin resistance, impairs lipid accumulation in adipose tissue, and increases hepatic steatosis in leptin-deficient mice. Diabetes, 2012. 61(12): p. 3208-18.
87.Zhang, Y.K., et al., Nrf2 deficiency improves glucose tolerance in mice fed a high-fat diet. Toxicol Appl Pharmacol, 2012. 264(3): p. 305-14.
88.Yu, Z., et al., Oltipraz upregulates the nuclear factor (erythroid-derived 2)-like 2 [corrected](NRF2) antioxidant system and prevents insulin resistance and obesity induced by a high-fat diet in C57BL/6J mice. Diabetologia, 2011. 54(4): p. 922-34.
89.Sampath, C., et al., Green tea epigallocatechin 3-gallate alleviates hyperglycemia and reduces advanced glycation end products via nrf2 pathway in mice with high fat diet-induced obesity. Biomed Pharmacother, 2017. 87: p. 73-81.
90.Kim, C.Y., et al., Parthenolide, a feverfew-derived phytochemical, ameliorates obesity and obesity-induced inflammatory responses via the Nrf2/Keap1 pathway. Pharmacol Res, 2019. 145: p. 104259.
91.Meher, A.K., et al., Nrf2 deficiency in myeloid cells is not sufficient to protect mice from high-fat diet-induced adipose tissue inflammation and insulin resistance. Free Radic Biol Med, 2012. 52(9): p. 1708-15.
92.Chartoumpekis, D.V., et al., Nrf2 deletion from adipocytes, but not hepatocytes, potentiates systemic metabolic dysfunction after long-term high-fat diet-induced obesity in mice. Am J Physiol Endocrinol Metab, 2018. 315(2): p. E180-E195.
93.Xue, P., et al., Adipose deficiency of Nrf2 in ob/ob mice results in severe metabolic syndrome. Diabetes, 2013. 62(3): p. 845-54.
94.Uruno, A., et al., Nrf2-Mediated Regulation of Skeletal Muscle Glycogen Metabolism. Mol Cell Biol, 2016. 36(11): p. 1655-72.
95.Chartoumpekis, D.V., et al., Nrf2 prevents Notch-induced insulin resistance and tumorigenesis in mice. JCI Insight, 2018. 3(5).
19. Line 226-227, "Two studies[74] [75] using…… needs reference formatting…o be merged
Thank you for bringing this to our attention. We have merged the references.
20. Line 245 what is Keap1 KD? is it knockdown? Need to expand KD atleast once
Thank you for your suggestion. We have added the abbreviation KD after the initial mention of knockdown.
21. The authors keep changing from Nrf2 KO to Nrf2 null mice? Need to keep one format throughout the article
Thank you for your insightful suggestion. We have changed all the Nrf2 null to Nrf2 KO to maintain consistency.
22. Line 257, in vitro and in vivo to be italics
Thank you for bringing this to our attention. We have italicized in vitro and in vivo in the manuscript.
Reviewer 3 Report
This review submitted by Li et al described the role of the Nrf2 signaling in obesity and insulin resistance. Nrf2 signal is known to be very complex and important to maintain homeostasis for normal function of several tissues. However, this is overall descriptive and lacks in-depth discussions. The first half is overlapped previous reviews, so the second half should be discussed more, for example cellular metabolism/immunometabolism and KEAP1-Nrf2 system in inflammation. And also, it is necessary to add some figures to help readers understand the contents.
Author Response
We thank the editor and reviewers for your thoughtful comments and positive feedback on our manuscript, which have helped us to improve the paper. We have revised the manuscript based on the reviewers’ suggestions and fixed the overlaps in the manuscript. Our point-to-point responses to individual comments are detailed below. We used the "track changes" function to highlight the changes we made in the resubmitted vision.
This review submitted by Li et al described the role of the Nrf2 signaling in obesity and insulin resistance. Nrf2 signal is known to be very complex and important to maintain homeostasis for normal function of several tissues. However, this is overall descriptive and lacks in-depth discussions. The first half is overlapped previous reviews, so the second half should be discussed more, for example cellular metabolism/immunometabolism and KEAP1-Nrf2 system in inflammation. And also, it is necessary to add some figures to help readers understand the contents.
Thank you for your invaluable insight. We have expanded the manuscript overall to better highlight the importance of inflammation and oxidative stress in insulin resistance and obesity and have also attempted to better explain the importance of Nrf2 system in inflammation. Additionally, we have also added a table for section 5.1 and 5.2 to organize the studies.
Round 2
Reviewer 1 Report
The authors addressed most of the comments revising the manuscript. The work allowed them to improve the overall complexity of the topic and, at the same time, to clarify confusing concepts present the first version of the paper. Indeed the quality of the actual version of the paper has greatly improved.
Reviewer 3 Report
If possible, I would suggest adding a summary figure for Nrf2 signaling.